# The aims and effectiveness of communities of practice in healthcare: A systematic review

**Alexander P. Noar** [ID][1,2¤a]*, **Hannah E. Jeffery**[3], **Hariharan Subbiah Ponniah** [ID][1], **Usman Jaffer** [ID][1,4¤b]

1 Department of Surgery and Cancer, Faculty of Medicine, Imperial College London, London, United Kingdom, 2 Highgate Mental Health Centre, Camden and Islington NHS Foundation Trust, London, United Kingdom, 3 Department of General Surgery, East and North Hertfordshire NHS Trust, Stevenage, United Kingdom, 4 Department of Vascular Surgery, Imperial College Healthcare NHS Trust, London, United Kingdom

¤a Current address: Highgate Mental Health Centre, London, United Kingdom
¤b Current address: St Mary's Hospital, London, United Kingdom
* a.noar@imperial.ac.uk

**Data Availability Statement:** All relevant data are within the paper.

**Funding:** The author(s) received no specific funding for this work.

## Abstract

Communities of practice (CoPs) are defined as "groups of people who share a concern, a set of problems, or a passion about a topic, and who deepen their knowledge and expertise by interacting on an ongoing basis". They are an effective form of knowledge management that have been successfully used in the business sector and increasingly so in healthcare. In May 2023 the electronic databases MEDLINE and EMBASE were systematically searched for primary research studies on CoPs published between 1st January 1950 and 31st December 2022. PRISMA guidelines were followed. The following search terms were used: community/communities of practice AND (healthcare OR medicine OR patient/s). The database search picked up 2009 studies for screening. Of these, 50 papers met the inclusion criteria. The most common aim of CoPs was to directly improve a clinical outcome, with 19 studies aiming to achieve this. In terms of outcomes, qualitative outcomes were the most common measure used in 21 studies. Only 11 of the studies with a quantitative element had the appropriate statistical methodology to report significance. Of the 9 studies that showed a statistically significant effect, 5 showed improvements in hospital-based provision of services such as discharge planning or rehabilitation services. 2 of the studies showed improvements in primary-care, such as management of hepatitis C, and 2 studies showed improvements in direct clinical outcomes, such as central line infections. CoPs in healthcare are aimed at improving clinical outcomes and have been shown to be effective. There is still progress to be made and a need for further studies with more rigorous methodologies, such as RCTs, to provide further support of the causality of CoPs on outcomes.

## Introduction

Medical knowledge is estimated to double every 73 days [1], leaving both physicians and patients with a seemingly insurmountable amount of information to stay on top of. This

**Competing interests:** Usman Jaffer is co-founder and CEO of health-shared.com an online platform that hosts communities of practice. This does not alter our adherence to PLOS ONE policies on sharing data and materials.

essentially means those involved in healthcare have to become skilled at knowledge management, defined as 'the collection of methods related to creating, sharing, using, and managing the knowledge and information of an organisation' [2].

One knowledge management strategy that has received significant attention is the theory of communities of practice (CoPs). CoPs are defined as "groups of people who share a concern, a set of problems, or a passion about a topic, and who deepen their knowledge and expertise by interacting on an ongoing basis" [3]. CoPs have a domain of interest, a community of individuals who all share that interest, and a practice consisting of the shared knowledge and skills built up by the community.

Initially described in the business sector, they have been particularly effective as a mechanism for the sharing of tacit knowledge [4]. First described by Polanyi, the Hungarian-British philosopher in 1966 [5], tacit knowledge, in comparison to explicit knowledge, is very difficult to directly codify and share in guidelines. It is best communicated through direct observation and imitation as well as through conversations, stories, and metaphors. The medical profession is a clear example of one where tacit knowledge is constantly used, exemplified by the 'mind-lines' (rather than guidelines) that practitioners tend to follow [6].

There has been an evolution of the concept, when initially described by Wenger and Lave, they were highly location specific, to a certain office or workspace, where individuals working together would interact, bouncing ideas off each other and helping newer members become fully integrated into the working environment. Over time, the description altered to include those who were not working together in the same physical place, but still shared the same domain of interest and were working on the same set of problems. This opened up the opportunity for virtual CoPs (vCoPs) to be included in the definition, where communities from all over the world interact digitally, producing the same tacit sharing effects as those working in the same physical space.

This review looks to elucidate the aims and effectiveness of CoPs in healthcare as well as communication methods used in these CoPs. We will also show what barriers and facilitators CoPs find when they are implemented in healthcare settings.

## Material and methods

In May 2023 the electronic databases MEDLINE and EMBASE were systematically searched for primary research studies on CoPs published between 1st January 1950 and 31$^{st}$ December 2022. PRISMA guidelines were followed.

The following search terms were used: community/communities of practice AND (healthcare OR medicine OR patient/s). The search was limited to research on human subjects and papers published in the English language. There was no restriction on geographical location.

This review was limited to original research with a focus on CoPs in the healthcare sector. Only papers published in peer- reviewed journals were included. Exclusion criteria were as follows:

- Studies reporting on CoPs in sectors other than healthcare.

- Studies reporting on medical education.

- Studies reporting on multiple interventions

- Case studies.

- Records with no abstracts.

- Study protocols

- Review articles

- News-style or opinion articles, theses and dissertations, and abstracts of conference proceedings without full peer-reviewed papers.

The search was completed using Ovid, and the reference list was uploaded to Covidence. Two authors (APN and HSP) independently reviewed all titles and abstracts, checking against inclusion and exclusion criteria. Relevant papers were marked for retrieval of full text and detailed review. When decisions differed, a final decision was made after discussion between the two reviewers. One author (APN) reviewed and extracted using a standardised template. Reference lists of included studies were also screened. When relevance of the paper was uncertain, or the findings were difficult to extract, APN discussed the paper with UJ. PRISMA flow diagram can be seen in Fig 1.

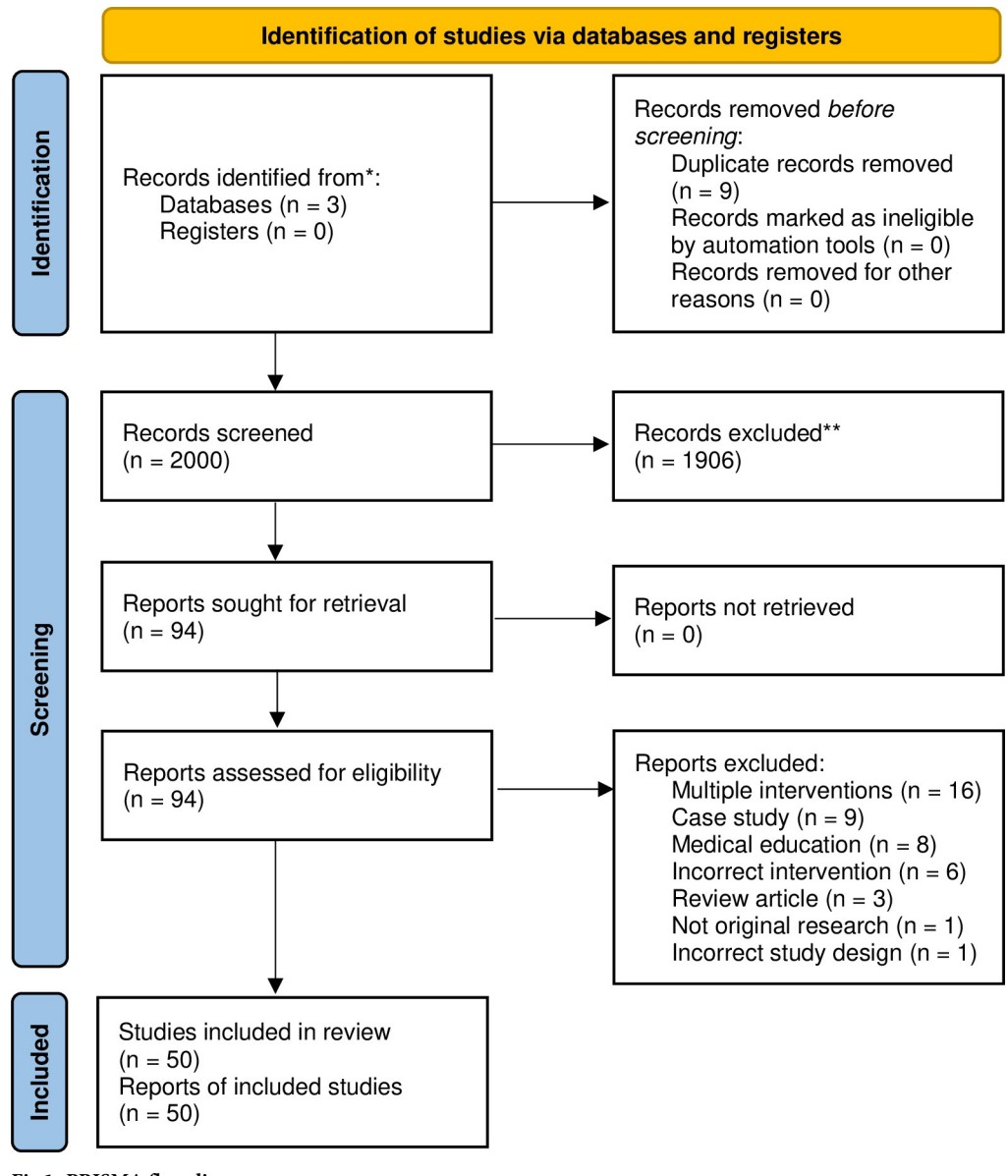

**Fig 1. PRISMA flow diagram.**

The following data were extracted: study details (author name, year of publication, country, sample size, study design, study type, data collection method, data analysis method, outcomes measured, barriers/facilitators, and limitations) and description of the CoP (including population, why it was established, how it was established, method of communication, and content shared).

Bias was assessed using the Critical Appraisal Skills Programme (CASP) checklist. Microsoft Excel was used to build tables of the studies included in this review. This review was not registered and a protocol was not prepared. Template data collection forms and data extracted from included studies is available upon request.

## Results

### Results

The database search picked up 2009 studies for screening, of which 94 studies were eligible for full-text review. Of these 50 papers met the inclusion criteria for this systematic review. The most frequent reason for exclusion at this stage was that the study included multiple interventions of which only one was a CoP. Total participants in CoPs across the studies were 12,400, with an average of 282 participants per study (6 studies did not report participant number).

### Country and year of publication

The most common frequent country that the studies were published in was Canada with 12 studies [7, 16, 19, 35, 38, 40, 41, 43, 45, 47, 54, 56], followed closely by the USA with 10 studies [9, 10, 14, 17, 23–25, 48, 51, 55], and the UK with 8 studies [8, 12, 13, 15, 18, 22, 31, 37]. Other notable contributions came from Australia with 6 studies [26, 29, 33, 34, 44, 50] and Spain with 4 studies [20, 27, 39, 42]. All other countries had 2 or less studies. As for year of publication, there was an overall trend of an increasing number of publications in more recent years. 2021 and 2015 had the largest number of studies with 7. 2019, 2018, 2016, and 2014 all have 4 studies. Only 2013, 2005, and 2007 had no studies published in those years.

### The aims of the CoPs

There were a number of themes that emerged from the aims of the CoPs examined in this study (Table 1). The most common by far was to directly improve a clinical outcome, with 19 studies aiming to achieve this. This included disease related factors such as reducing central line infections [9], improving glucose control in critically ill patients [40], and increasing viral suppression rates in HIV [51]. This theme also included many aspects of improving clinical services and workflows such as improving rehabilitation for patients with AF [49], improve pain practices for spinal cord injury patients [43], and improve the falls prevention care for care-home residents [34].

Developing skills was also a common reason for setting up a CoP with 8 studies in this theme. This included building research skills [29, 32] and developing self-care techniques [48]. There were also 7 studies whose aim was to share best-practice. This included the direct sharing of evidence-based practice [12, 50] as well as trying to decrease variation in practice over geographically spread out areas by providing clinicians in the same speciality a means of communication [11, 41, 44].

Sharing specialist knowledge was the aim of 6 studies. Of these, 4 were aimed at connecting primary care physicians with hospital-based specialists [14, 27, 39, 45] for example providing rural primary care physicians the knowledge to manage patients with chronic hepatitis C

**Table 1. Aims and effectiveness of included studies.**

| Ref | Bias | Year of Publication | Authors | Location | Participants | Study Design | Outcome Measure | Aim | Effectiveness |
|---|---|---|---|---|---|---|---|---|---|
| [7] | High | 2003 | Gagliardi et al. | Canada | 22 | Non-randomised experimental study | Mixed | To facilitate interaction between community-based general surgeons and oncologists in a tertiary care setting through interactive multidisciplinary rounds. | Feasible to engage remote surgeons in multidisciplinary oncology rounds by videoconference. 25% of participants said that their practice would change. |
| [8] | Med | 2004 | Russel et al. | UK | 2800 | Qualitative Research | Qualitative | To promote evidence based healthcare by linking practitioners with researchers | Communities of practice emerged from the informal email network. The network helped to bridge the gap between research and practice providing the opportunity to collaborate across boundaries. |
| [9] | Med | 2006 | Render et al. | USA | / | Non-randomised experimental study | Mixed | To reduce the number of central line infections in hospitals | All sites reduced central line infections by 50% (1.7 to 0.4/1000 line days, p<0.05). Adherence to evidence based practices increased from 30% to nearly 95%. |
| [10] | Low | 2008 | White et al. | USA | 74 | Qualitative Research | Qualitative | To enhance quality of care and safe practices in acute and community care departments in a rural hospital | CoPs enhanced interprofessional practice through improving communications, such as introducing joint care meetings, or information transfer, such as streamlining discharge processes. |
| [11] | Med | 2008 | Falkman et al. | Sweden | 24 | Qualitative Research | Mixed | To improve the ability of oral medicine to share cases and learn from each other due to their geographically dispersed speciality. | The introduction of SOMWeb improved the structure of meetings and their discussions, and a tenfold increase in the number of participants. The platform has been adopted as the national website for continuing education in oral medicine. |
| [12] | Low | 2008 | Tolson et al. | UK | 24 | Non-randomised experimental study | Mixed | To promote evidence-based practice in NHS sites | 80% of patient related criteria and 35% of the facilities criteria were achieved. The Revised Nursing Work Index indicated the nurses experienced greater autonomy (p = 0.019) and increased organisational support (p = 0.037). |
| [13] | Low | 2009 | Griffiths et al. | UK | 19 | Qualitative Research | Qualitative | To satisfy the workplace demands that the nurses faced on medical assessment units | The main themes identified regarding the nurses role were organising the clinical space, having professional knowledge, and having the ability to work under pressure. |

*(Continued)*

**Table 1.** (Continued)

| Ref | Bias | Year of Publication | Authors | Location | Participants | Study Design | Outcome Measure | Aim | Effectiveness |
|---|---|---|---|---|---|---|---|---|---|
| [14] | Med | 2010 | Arora et al. | USA | / | Qualitative Research | Mixed | To develop knowledge and skills in provincial primary care providers regarding management of hepatitis C virus | Clinicians report increased competence in all nine abilities for HCV management after 12 months of participation e.g. ability to treat patients with HCV and manage side effects Likert scale average 2.0 to 5.2 (p<0.0001). 98% of respondents thought that ECHO participation had either a moderate or major benefit on enhancing knowledge about management and treatment of patients with HCV. Clinical providers found the case-based learning the most essential source of learning. |
| [15] | High | 2010 | Skirton et al. | UK | 156 | Qualitative Research | Qualitative | To develop standards and a code of practice for genetic counselling to guide professionals in Europe. | The members of the CoP developed a set of professional standards and a code of practice. Suggestions included making genetic counsellor a protected title requiring a master level degree in genetic counselling. |
| [16] | Med | 2011 | Burgess et al. | Canada | 11 | Qualitative Research | Qualitative | To engage nurse practitioners in social investigation, education and actions, and to explore how collaboration advances their role in primary healthcare | CoP helped NPs to build collaborative relationships, enhance practice learning and competence, extend and apply new knowledge, enrich professional identities, and shape health organisational policy and politics. CoP is seen as a major factor for the 100% retention rate of NPs. CoP facilitated exchange of ideas that led to many successful abstract submissions. Participation in the CoP helped build a better sense of the unique identity of being a NP. |
| [17] | High | 2011 | Massett et al. | USA | / | Non-randomised experimental study | Quantitative | To help with the issues of oncology clinical trial accrual | AccrualNet has had more than 45000 views, with the Tools and Resources, Conversations, and Training sections being the most viewed. Total content has increased by 69%. Total conversations were 29 with 43 posts. |

*(Continued)*

**Table 1.** (Continued)

| Ref | Bias | Year of Publication | Authors | Location | Participants | Study Design | Outcome Measure | Aim | Effectiveness |
|---|---|---|---|---|---|---|---|---|---|
| [18] | Med | 2013 | Adams et al. | UK | 44 | Qualitative Research | Qualitative | To facilitate informal learning amoung nurses in community services | The higher performing service (service B) had more time for catch ups in comparison to the lower performing service (service A). An erosion of workplace relationships left them feeling alone and unsupported in service A. Service B phoned around so many nurses went to lunch at the same time. The ideas discussed during catch ups helped staff develop a better understanding of approaches to patient care. |
| [19] | High | 2014 | Fung-Kee-Fung et al. | Canada | 230 | Non-randomised experimental study | Quantitative | To improve cancer care in a regional quality improvement collaborative. | The CoP aided development of a collaboration between hospitals that saw compliance with guidelines improve by 20%, as well as the standardisation of peri-operative pathways in a number of disease sites. Increases in the use of sentinel lymph node biopsy in breast cancer surgery and decreased positive surgical margin rates in prostate cancer were also seen. |
| [20] | Med | 2014 | Diaz-Chao et al. | Spain | 169 | Non-randomised experimental study | Quantitative | To improve primary care and reduce hospital referrals. | Use of the platform improved primary care ($p<0.001$) and led to fewer hospital referrals ($p<0.05$). When healthcare staff used social networks and ICT technologies professionally, and had more contact hours with patients, the more the platform was used for communication between primary and hospital care professionals. |
| [21] | Med | 2014 | Bindels et al. | Netherlands | 13 | Qualitative Research | Qualitative | To evaluate the implementation of programs that provide care for frail older people | CoP members had unanticipated concerns regarding the pro-active approach of the programs and older people not being open to receiving care. CoP is a useful strategy as part of an evaluation aimed at improving program implementation. CoP allowed for moral issues of providing care, such as care avoidance, to be discussed, for which there are no guides of how to manage. CoP created a social infrastructure, which allowed for more collaboration. |

*(Continued)*

**Table 1.** (Continued)

| Ref | Bias | Year of Publication | Authors | Location | Participants | Study Design | Outcome Measure | Aim | Effectiveness |
|---|---|---|---|---|---|---|---|---|---|
| [22] | Med | 2014 | Carolan et al. | UK | 43 | Qualitative Research | Qualitative | To help parents of children and young people with CKD engage in an online platform to aid shared responsibility for condition management. | Evolving communities of child-healthcare practice were identified comprising three components: Parents making sense of clinical tasks, parents executing tasks according to their individual skills, and parents defining task and group members' worth and creating a personal identity within the community. |
| [23] | Med | 2015 | Meins et al. | USA | 58 | Non-randomised experimental study | Mixed | To provide specialist pain management consultation to community healthcare providers without access to these services locally. | Telepain was determined to be a CoP by displaying the 14 indicators of a CoP described by Wenger. Telepain also enhanced the knowledge of community healthcare provider's regarding pain management strategies (average score 3.94/4) as well as increasing their confidence (3.77/4). |
| [24] | Low | 2015 | Shaikh et al. | USA | 31 | Qualitative Research | Qualitative | To increase assessment and counseling for childhood obesity prevention | The main challenges to the quality improvement project HEALTH COP were getting staff buy-in, changing ingrained clinical practices, and motivating patients and families. Facilitators were top down requirements for QI, linkages to QI resources, involvement of clinical champions, alignment with existing practices, incorporating a learning system connecting similar clinics, and clear communication channels. |
| [25] | Med | 2015 | Heidenreich et al. | USA | 305 | Randomised Controlled Trial | Quantitative | To aid the enrolment and adoption of the Hospital to Home quality improvement initiative to improve the transition of care for hospitalised patients with heart disease. | 54% of hospitals randomised to the CoP intervention arm enrolled patients into Hospital to Home (H2H), compared to 10% in the control arm (p<0.001). Intervention hospitals had more ongoing or planned projects related to H2H (p<0.001). Total cost of CoP facilitation was estimated at $10,200. |
| [26] | Med | 2015 | Jefford et al. | Australia | / | Qualitative Research | Qualitative | To trial novel models of post treatment care in cancer patients | Cancer patients found the interventions to be acceptable, appropriate, and effective. |

*(Continued)*

**Table 1.** (Continued)

| Ref | Bias | Year of Publication | Authors | Location | Participants | Study Design | Outcome Measure | Aim | Effectiveness |
|---|---|---|---|---|---|---|---|---|---|
| [27] | Low | 2015 | Lacaster Tintorer et al. | Spain | 166 | Qualitative Research | Mixed | To improve the communication between primary care and specialist healthcare professionals. | The most important factor for engagement with the CoP was the perceived usefulness for reducing costs of clinical practice. Both perceived usefulness for improving the quality of clinical practice and habitual social media use also helped to drive engagement. |
| [28] | Med | 2015 | Dong et al. | International | 500 | Qualitative Research | Mixed | To aid hand surgeons with continuing professional development | Number of members grew from 38 to 4106. Members perceived the LinkedIn community as user-friendly and easy to use. 42% answered strongly agree, and 37% agree to the question 'How would you rate the overall ease of using the platform?'. System usability scale score 84.6. |
| [29] | Med | 2016 | Gullick et al. | Australia | 25 | Qualitative Research | Qualitative | To build research skills for nurses in busy clinical environments. | The CoP created enduring research relationships and participants described significant value to the research culture that was developed. Many examples of research dissemination and enrolment in doctoral programmes came from participation in the CoP. |
| [30] | Low | 2016 | McCreesh et al. | Ireland | 12 | Qualitative Research | Qualitative | To help physiotherapists working in primary care manage shoulder pain | A desire for peer supports was the strongest motivator for joining. Barriers including not having enough time to engage fully due to work pressures. The access to meetings, the provision of preparation work, and deadlines for the journal clubs were reported as facilitators. Benefits included reported positive clinical practice changes as well as personal growth and development particularly with evidence-based practice skills. |
| [31] | Low | 2016 | Wallis et al. | UK | 26 | Qualitative Research | Qualitative | To improve the management of TB | Participants described the development of a community of practice. The audit promoted local and regional team working, exchange of good practices, and local initiatives to improve care. |

(*Continued*)

**Table 1.** (Continued)

| Ref | Bias | Year of Publication | Authors | Location | Participants | Study Design | Outcome Measure | Aim | Effectiveness |
|---|---|---|---|---|---|---|---|---|---|
| [32] | Med | 2016 | Becerril-Montekio et al. | Mexico | 200 | Qualitative Research | Qualitative | To strengthen healthcare professionals capacities to acquire, analyse, adapt, and apply research results. | Quality of healthcare was seen as the most important problem of the state departmental health system that represents an obstacle to reach the expected results of maternal health programs. Quality of healthcare and excess of patient demand were seen as the most feasible problems to solve. |
| [33] | Low | 2016 | Terp et al. | Australia | 11 | Qualitative Research | Qualitative | To co-design a smartphone application for use in early schizophrenia care. | The major categories supporting an engaging environment were: a pre-narrative about a community of practice; the room for design is a community of practice; and the community of practice as a practice of special qualities. Participatory design can support and inspire participation and engagement in the development of mental health care with young adults with schizophrenia. |
| [34] | High | 2017 | Francis-Coad et al. | Australia | 20 | Qualitative Research | Mixed | To help reduce the number of falls in residential aged care sites. | The audit conducted by the CoP revealed gaps in practice such as the number low number of residents receiving Vitamin D, the lack of a mandatory falls prevention education for staff, and no falls prevention policy. Actions included requesting that GPs prescribe vitamin D, defining falls, and writing a falls prevention policy. |
| [35] | Med | 2017 | Camden et al. | Canada | 41 | Non-randomised experimental study | Mixed | To improve physical therapists' self-perceived practice | Self-perceived knowledge, skills, and practice change scores were significantly higher (+0.47, +1.23, and +2.61 respectively; p<0.001) at the end of the CoP compared with the beginning. CoP also significantly impacted belief about capabilities and social influence (+6.64 p<0.002, +5.08 p<0.03 respectively). |

(*Continued*)

**Table 1.** (Continued)

| Ref | Bias | Year of Publication | Authors | Location | Participants | Study Design | Outcome Measure | Aim | Effectiveness |
|---|---|---|---|---|---|---|---|---|---|
| [36] | Med | 2018 | Cheng et al. | International | 688 | Qualitative Research | Mixed | To encourage collaborative, multi-centre simulation-based research. | The network successfully completed and published numerous collaborative research projects in simulation. INSPIRE has won grant funding for infrastructure support. All 14 of Wenger's indicators for the presence of a community of practice were found. |
| [37] | Low | 2018 | Weiringa et al. | UK | / | Qualitative Research | Qualitative | To allow physicians to discuss patient care and share experiences | Very few posts in the virtual communities of practice referred to explicit guidelines. Instead individual cases highlighted outliers. Tacit, rather than explicit, knowledge was expressed as well as pragmatic reasoning focusing on particular cases. Discussion were reinforced through stories, jokes, and imagery. |
| [38] | High | 2018 | Fingrut et al. | Canada | 148 | Non-randomised experimental study | Quantitative | To decrease barriers to access, foster collaboration, and improve knowledge of guidelines in cancer care. | Participants mostly agreed or strongly agreed that the CoP reduced barriers (76.0%), improved access (82.4%), fostered teamwork (84.5%), improved knowledge (93.3%), improved standards of practice (92.3%), and increased satisfaction in caring for patients (82.9%). The CoP also brought members of the government and hospital administration together with frontline clinicians. |
| [39] | Low | 2018 | Lacaster Tintorer et al. | Spain | 29 | Qualitative Research | Qualitative | To facilitate the communication between primary care and specialist healthcare professionals. | Participants reported that the tool should be integrated into habitual clinical workstations to be of most effect. They also thought contact with specialists should be virtual and that they should be provided with specific time to access the tool. |
| [40] | Med | 2019 | Dodek et al. | Canada | 272 | Non-randomised experimental study | Quantitative | To improve glucose control in critically ill patients | No significant changes to the average hyperglycaemic index, hypoglycaemic events, or standardised mortality rate in response to interventions. |

(*Continued*)

**Table 1.** (Continued)

| Ref | Bias | Year of Publication | Authors | Location | Participants | Study Design | Outcome Measure | Aim | Effectiveness |
|---|---|---|---|---|---|---|---|---|---|
| [41] | Med | 2019 | Glicksman et al. | Canada | 275 | Non-randomised experimental study | Mixed | To rebuild the provincial radiation therapy community to facilitate collaboration among centres, with the aim of decreasing variation in practice. | 95% of participants reported that CoP projects were very relevant to them, and 50% reported changes in their practice due to the CoP. 90% reported growth in their professional network and 93% felt the CoP was worthwhile. |
| [42] | Med | 2019 | Bermejo-Caja et al. | Spain | 12 | Qualitative Research | Qualitative | To improve the attitude of primary care professionals to the empowerment of patients with chronic conditions | GPs found the vCoP useful as it could provide up to date resources that could be used at the point of care. Both professionals found that discussing experiences with others helped them consider alternative approaches and advance learning. |
| [43] | High | 2019 | Savoie et al. | Canada | 77 | Non-randomised experimental study | Quantitative | To improve pain practices for spinal cord injury patients | Adherence to pain best practices for SCI exceeded 70% for most outcomes, all of which were improvements on the retrospective cohort. This included improvements in developing inter-professional pain treatment plans from 12% to 74%, and documenting pain onset from 4.5% to 80%. |
| [44] | Low | 2020 | Rolls et al. | Australia | 133 | Qualitative Research | Mixed | To facilitate communication and knowledge sharing between the clinicians working at the 43 adults ICUs in New South Wales | Nurses contributed 68% of posts and physicians 27%. Knowledge supplied was either experiential (35%), explicit (17%), both (17%), know-how (20%), know-why (5%), or no-knowledge exchanged (6%). Three subject areas were identified: clinical practices (71%); equipment (23%); and clinical governance (6%). Six elements facilitated participation and knowledge exchange: discussion thread, sharing of artefacts, community, cordiality, maven work, and promotion of the community. |
| [45] | High | 2020 | Pariser et al. | Canada | 616 | Qualitative Research | Mixed | To provide streamlined access to specialist care and virtual-team based resources for primary care. | A CoP was successfully formed between primary care and specialist care. This also led to new initiatives being created that responded to primary care needs, such as facilitating real time access to radiology services. These initiatives led to a perceived reduction in ED visits by 40%. |

*(Continued)*

**Table 1.** (Continued)

| Ref | Bias | Year of Publication | Authors | Location | Participants | Study Design | Outcome Measure | Aim | Effectiveness |
|---|---|---|---|---|---|---|---|---|---|
| [46] | Low | 2020 | McCurtin et al. | Ireland | 15 | Qualitative Research | Qualitative | To encourage clinician research engagement by linking them with researchers in higher educations institutions | Members of the CoP felt the priorities (in order) of the CoP should be: dissemination, education, enablers, networking, and advocacy. Actions proposed included the development of a research database, to act as advocates, as well as lobbying for clinical-research posts. |
| [47] | High | 2021 | Hahn-Golberg et al. | Canada | / | Non-randomised experimental study | Mixed | To implement the patient orientated discharge summary | High participation in the community of practice was associated with higher penetration. 64% of patents across the hospitals received a patient orientated discharge summary (PODS). PODS improved family involvement during discharge teaching (7% increase. p = 0.026). |
| [48] | Med | 2021 | Katzman et al. | USA | 1530 | Non-randomised experimental study | Mixed | To provide education for first responders on self-care techniques and stress resilience. | Overall stress levels did not decline, but participants felt more confident in using psychological first aid, managing others who needed mental health assistance, and taking time for self-care. They also had a significant reduction in how isolated they felt. |
| [49] | Med | 2021 | Dinesen et al. | Denmark | 20 | Non-randomised experimental study | Mixed | To improve rehabilitation of patients with AF | Patients found the program useful and felt more secure living with AF. Patients also displayed increased knowledge about AF at follow-up compared with baseline (p = 0.02). |
| [50] | High | 2021 | Keir et al. | Australia | 3228 | Non-randomised experimental study | Quantitative | To facilitate the spread of information regarding neonatal evidence based medicine | Since the registration of the hashtag, it has been used in 23939 tweets and 37259710 impressions were generated. The majority of users made one tweet using the hashtag (n = 1078), followed by two tweets (n = 411), and more than 10 tweets (n = 347). The online community contained the critical components of a community of practice. |
| [51] | Med | 2022 | Steinbock et al. | USA | 90 | Non-randomised experimental study | Quantitative | To increase viral suppression rates in populations disproportionately affected by HIV | The average viral suppression rates for the selected populations increased from 79.2% to 82.3%. The viral suppression gap between the selected disadvantaged groups and the rest of the served HIV population was reduced from 5.7% to 3.8%, a 33.5% reduction. |

(*Continued*)

**Table 1.** (Continued)

| Ref | Bias | Year of Publication | Authors | Location | Participants | Study Design | Outcome Measure | Aim | Effectiveness |
|---|---|---|---|---|---|---|---|---|---|
| [52] | Low | 2021 | Gerritsen et al. | Netherlands | 101 | Qualitative Research | Qualitative | To support the implementation of the psychiatric intensive care approaches. | Key insights included the need to create an ambassador role for CoP participants, to organise concrete activities, be mindful of the multi-disciplinary composition, to foster shared responsibility, and to work on sustainability. The CoP was perceived to help support and further develop the HIC and FHIC approaches. |
| [53] | Low | 2022 | Montali et al. | Italy | 16 | Qualitative Research | Qualitative | To give breast cancer patients a space to talk about their experiences and receive peer support. | Analysis revealed five processes that breast cancer patients go through including: mirroring, monitoring, modelling, belonging, and distancing. The community contributed to the participants' sense of empowerment. |
| [54] | Low | 2022 | Dames et al. | Canada | 94 | Non-randomised experimental study | Mixed | To deliver a 12 week ketamine-assisted therapy program | Pre post scores: PHQ-9 13 (moderate) to 7 (mild), PCL-5 47 (moderate) to 20 (mild), GAD-7 12 (moderate) to 6 (mild), B-IPF 42 (moderate) to 18 (mild). 91% of GAD and 79% of depression went into a milder category. 86% of PTSD screen negative and 92% of those with life work impairments had significant improvements. |
| [55] | Low | 2022 | Rushanan et al. | USA | 13 | Non-randomised experimental study | Mixed | To build the competence of occupational therapists treating patients with neurodegenerative diseases | The clinical competency assessment tool for occupational therapists treating patients with neurodegenerative diseases (CAT) for knowledge improved from 26.9 to 35.7 (p = 0.002), for beliefs improved from 28.7 to 35.2 (p = 0.001), and for actions improved from 25.2 to 31.9 (p = 0.002). |
| [56] | Med | 2022 | Sibbald et al. | Canada | 17 | Non-randomised experimental study | Mixed | To connect mid-career professionals from across Canada who are committed to improving healthcare police and practice | The program was successful in helping participants make connections (mean = 2.43). Participants reported the development of a sense of belonging (mean = 2.29) and facilitated knowledge exchange (mean = 2.43). At the time of this study, participants felt the program had minor impact on their work (mean = 3.5). |

infection [14]. Another 3 studies brought clinicians together with researchers with the aim to stimulate research ideas and activity [8, 36, 46].

Other notable CoPs were set up with the specific aim to complete a specific task, such as develop a set of standards for genetic counselling in Europe [15], or to co-design a smartphone application with patients for schizophrenia care [33].

## Effectiveness of the CoPs

The effectiveness of the CoPs was measured in a variety of ways (Table 1). 30 studies were qualitative research, 20 studies used a non-randomised experimental design, and 1 study was a randomised controlled trial [25]. In terms of outcomes, qualitative outcomes were the most common measure used in 21 studies, a mix of both qualitative and quantitative outcomes were used in 20 studies, and solely quantitative outcomes were used in 9 studies. Only 11 of the studies with a quantitative element had the appropriate statistical methodology to report significance. All except 1 study [40] reported a positive significant effect when implementing a CoP. Outcomes varied across geographical location with North American countries such as Canada (91.7%) and USA (80%) having a higher percentage of studies with a quantitative element to their outcomes, in comparison to the UK (12.5%) or Australia (50%).

Of the 9 studies that showed a statistically significant effect, 5 showed improvements in hospital-based provision of services [12, 25, 35, 47, 49, 55]. These studies included implementing patient orientated discharge summaries leading to an 7% increase (p = 0.026) of family involvement during discharge [47], as well as another study improving rehabilitation services for patients with atrial fibrillation (AF) which demonstrated an increase in patients' knowledge about AF (p = 0.02) [49]. 2 of the studies showed improvements in primary-care. Arora et al. showed how bringing primary care providers together with hospital specialists improved primary care knowledge about the management of hepatitis C infection (p<0.0001). Diaz-Chao et al. showed how bringing primary care physicians together with specialists led to fewer hospital referrals (p<0.05). Finally, 2 studies showed improvements in direct clinical outcomes. One study showed a reduction in central line infections by 50% (p<0.05) (9) and another showed an increase in HIV viral suppression rates from 79.2% to 82.3% (p<0.05) [51].

## Communication

Table 2 describes the methods of communication utilised by each of the communities of practice described in the 50 studies included in this review. Of the communities of practice 23 communicated virtually, 12 communicated face-to-face, and 13 used both face to face and virtual methods of communication. In two of the studies [26, 41], it is unclear whether the communication was virtual, face-to-face or both. 23 of the communities of practice held meetings for the members, 10 utilised workshops, 8 described seminars, and 1 described tutorials [42]. 25 studies communicated using web-based systems and blogs and 10 communicated via email. 18 of the studies described other methods of communication, which included video consultation [49], telephone-based catch-ups [14, 18] and case based presentations/discussions [14, 19, 23, 51]. The average year for face-to-face only communication was 2014.25 (SD 4.94) and 2015.78 (SD 5.56) for virtual only communication, which was not significantly different (p = 0.43).

## Barriers and facilitators

Barriers to engagement were reported in 15 of the studies; examples are given in Table 3. The biggest barrier to engagement was time constraints, reported in nine of the studies. Lack of space to meet up [18], or to access the vCOP [42] was reported in two of the studies, and lack of funding [43] or resource constraints [12] as a barrier was reported in two studies. Difficulty

**Table 2. Methods of communication.**

| Ref | Face-to-Face | Virtual | Workshops | Seminars | Meeting of Members | Emails | Web Based Systems and Blogs | Other |
|---|---|---|---|---|---|---|---|---|
| [7] | | Yes | | Yes | | | | |
| [8] | | Yes | | | | Yes | | Personalised targeting of content based on interests |
| [9] | Yes | | Yes | | Yes | | | Presentation from the monthly members meeting posted on bulletin boards. |
| [10] | Yes | | | | Yes | | | |
| [11] | | Yes | | | | Yes | Yes | |
| [12] | Yes | Yes | | | Yes | | Yes | |
| [13] | Yes | | | | | | | Organically working together on the ward |
| [14] | | Yes | | | | | | Weekly 2hr telemedicine clinics |
| [15] | Yes | Yes | | | Yes | Yes | Yes | |
| [16] | Yes | | | | Yes | | | |
| [17] | | Yes | | | | | Yes | |
| [18] | Yes | | | | | | | Over the phone catch-ups |
| [19] | Yes | Yes | | | | | | Case-conferences |
| [20] | | Yes | | | | Yes | Yes | Document and image repository |
| [21] | Yes | | | | Yes | | | |
| [22] | | Yes | | | | | | |
| [23] | | Yes | | Yes | | | | Case-based discussions |
| [24] | | Yes | | | Yes | | | |
| [25] | | Yes | | Yes | | Yes | Yes | |
| [26] | | | | | Yes | | | |
| [27] | | Yes | | | | Yes | Yes | Document and image repository |
| [28] | | Yes | | | | | Yes | |
| [29] | Yes | Yes | | | | | Yes | |
| [30] | Yes | Yes | | Yes | Yes | | Yes | Journal club |
| [31] | Yes | | | | Yes | | | |
| [32] | Yes | Yes | Yes | | | | Yes | |
| [33] | Yes | | Yes | | Yes | Yes | | |
| [34] | Yes | Yes | Yes | | | | Yes | |
| [35] | Yes | Yes | Yes | | | | Yes | |
| [36] | Yes | Yes | | | Yes | | Yes | Speed dating, keynote speaker, and meeting feedback |
| [37] | | Yes | | | | | Yes | |
| [38] | Yes | | | | Yes | | | |
| [39] | | Yes | | | | Yes | Yes | |
| [40] | Yes | Yes | | Yes | | | Yes | Critical care quality day |
| [41] | | | | | | | | |
| [42] | | Yes | | Yes | | | Yes | |
| [43] | Yes | | Yes | | Yes | | | |
| [44] | | Yes | | | | Yes | | |
| [45] | Yes | Yes | Yes | | Yes | Yes | Yes | |
| [46] | Yes | | Yes | | Yes | | | |
| [47] | | Yes | | | Yes | | Yes | Mentorship |
| [48] | | Yes | | Yes | Yes | | | Weekly learning-listening sessions |
| [49] | Yes | Yes | | Yes | | | Yes | |
| [50] | | Yes | | | | | Yes | |
| [51] | | Yes | | Yes | Yes | | Yes | Case presentations |
| [52] | Yes | | Yes | | Yes | | | |

(*Continued*)

**Table 2.** (Continued)

| Ref | Face-to-Face | Virtual | Workshops | Seminars | Meeting of Members | Emails | Web Based Systems and Blogs | Other |
|---|---|---|---|---|---|---|---|---|
| [53] | | Yes | | | | | Yes | |
| [54] | Yes | Yes | | | Yes | | | |
| [55] | | Yes | | | Yes | | Yes | |
| [56] | | Yes | | | Yes | | | |

accessing the COP platform via usual workstations [39] or operating systems [42] was listed as a barrier in two of the studies. A lack of understanding of the concept of the COP was reported as a barrier in one study [10]; two studies cited fear of judgement as barriers to engagement [11, 53]. One study noted that those who were encouraged to join the COP by peers had lower engagement than those who self-selected [29], whilst another found that lack of participation by peripheral members caused frustration among core members [21].

Facilitators were reported in 24 of the studies; examples are given in Table 4. The most commonly highlighted facilitators were involvement of key members of the team. Primarily these were clinical, with studies citing strong clinical leadership [26, 39], support from health [16] or hospital [9] leadership, clinical champions [11, 24], experts [34, 38], and involving PCPs in the early stages of development [45]. Non-clinical roles were also highlighted, with two of the studies listed having a group facilitator important for engagement [21, 42], and one highlighting the importance of funding for an administrative coordinator [36]. One study found that a mentoring scheme helped to distribute expertise [36], whilst another found the opportunity for new members to learn through passive participation to be a facilitator [8]. Regular face to face meetings were listed by three of the studies as facilitators [36, 38, 49], with one study noting that using face-to-face and virtual activities supported different learning styles [35]. Use of social networks and ICT technologies in professional practice were found to be facilitators in one of the studies [36]. Alignment with existing practices, in particular with quality improvement methodology, was noted to be a facilitator in two of the studies [24, 29].

# Discussion

This systematic review has elucidated the aims and effectiveness of CoPs established in a healthcare setting. As described above, there were a variety of aims for the CoPs, with the majority relating to improving clinical outcomes and knowledge. Although encouraging to see the focus of these CoPs on clinically relevant issues, there were only 3 studies [9, 40, 51] where

**Table 3. Barriers to engagement.**

| Barrier | Example | Ref |
|---|---|---|
| Time constraints | The time commitment was the biggest barrier | [41] |
| Space constraints | Barriers included. . . a lack of space to meet up | [18] |
| Resource constraints | A lack of funding resulted in longer implementation times | [43] |
| Information Technology constraints | Not having the tool integrated into usual work stations. . . proved to be a barrier | [39] |
| Lack of understanding | Barriers included not understanding the CoP concept | [10] |
| Fear of judgement | Barriers. . . included. . . concern about how interesting a case is, and showing a gap in one's knowledge | [11] |
| Mode of selection | Those who were encouraged to join the CoP by peers, rather than self-selecting, had lower engagement | [29] |

**Table 4. Facilitators of engagement.**

| Facilitator | Example | Ref |
|---|---|---|
| Clinical leadership | Strong clinical leadership was the most important success factor | [26] |
| Hospital/health leadership support | Support of the CoP by health leaders was a major facilitator | [16] |
| Expert knowledge | Facilitators included access to a panel of experts | [34] |
| Group facilitators | The facilitator motivated members to contribute and filtered in relevant information | [42] |
| Clinical champions | Facilitators included the existence of a champion in the field | [11] |
| Administrative coordinator | The funding for the administrative co-ordinator has been a facilitator | [36] |
| Quality improvement methodology | Methodology that closely resembled quality improvement and allowed for quick wins kept the groups engaged | [29] |

the outcome measurement was a patient related clinical outcome with the suitable statistical methodology to determine a significant effect.

Furthermore, only 1 study [25] had a randomised control trial (RCT) design and therefore the ability to establish causality. In this study, 122 veterans affairs (VA) hospitals were randomised to have enrolment into a new initiative facilitated either by a CoP or through usual means—the standard national announcements that all hospitals receive for new initiatives. The initiative was the national hospital to home (H2H) project, and uptake to the programme was the primary outcome measure. H2H aimed to help inpatients with heart disease transition back to their place of residence through measures such as early follow up and patient education to recognise early signs of deterioration. The CoP was an already existing entity that had been set up previously to connect VA hospitals to improve the quality of care for patients with heart disease. The primary means of communication of the CoP was via email and they also had bimonthly teleconferences. 54% of the hospitals randomised to CoP facilitated arm enrolled in the H2H initiative whereas only 10% of those not facilitated by the CoP enrolled (p < 0.001). This is clear evidence of the effectiveness of utilising CoPs, albeit indirectly, for changing clinical practice. However, the ultimate goal of the H2H was to reduce 30-day readmission rates by 20%, and this study did not measure and compare this, which would have provided a more clinically meaningful endpoint.

Although not formally described as CoPs, and therefore were not picked up in the systematic search of this review, there are other RCTs published in the literature that provide support for the effectiveness of CoPs directly on clinical outcomes. Described as peer-mentoring schemes or online communities, lifestyle interventions that bring patients together who share the same set of problems, such as poor glucose control or low activity levels, have been highly effective at motivating patients to alter their behaviour [57, 58]. Richardson et al. conducted an RCT that provided the intervention arm with means of communication with their fellow patients during an online intervention to increase physical activity. Those able to communicate with their fellow participants, through posting and reading messages on a web-based blog, had a significantly reduced attrition rate than those who had no means of communication (79% v 56%, p = 0.02).

As the most common outcome of the CoPs was a change in practice, it is clear that as well as being a knowledge management strategy, CoPs are also behaviour change interventions. The capability, opportunity, and motivation model of behaviour change (COM-B) is a systematic way of framing the different facets required for an individual to change their behaviour [59]. Capability is defined as the psychological and physical requirements to perform the task. Opportunity represents the physical and social factors outside of the individual that make the behaviour possible, and motivation is defined as both the reflective and automatic brain

activity that energises and directs behaviour. Through the community and shared problem solving that CoPs offer, it is clear that they provide individuals with the psychological capability, social opportunity, and motivation they need for behaviour change through the learning resources and peer support available.

The main barrier to engagement was time constraints, which are to be expected in an overwhelmed healthcare environment that is busier than ever [60]. Funding constraints were also noted, which once again is not a surprise as healthcare spending as a percentage of GDP is at its lowest in a decade [61]. It is, however, encouraging to see that there were no barriers relating to a lack of digital skills, despite many individuals known to struggle [62]. With the digital revolution taking place in healthcare, strong digital skills in the workforce will be necessary to control spiralling costs. Such skills will be necessary for vCoPs to be taken up in a meaningful way across the healthcare ecosystem.

The main facilitator for engagement was strong leadership, including support from institutional leaders, which represents an alteration from the original vision for CoPs as self-organising entities with a lack of centralised leadership. This shows the specific healthcare related factors that many interventions face in the highly regulated and controlled environment. Future CoP endeavours should bear this in mind and make sure support and buy in is gained from the relevant clinical and administrative leaders. This will also help alleviate the main barrier to engagement by providing support or even specific protected time for the CoP.

CoPs differ from other knowledge management strategies such as work groups or knowledge networks. In work groups goals are pre-determined by a manager and members are usually assigned or selected by a leader. CoPs on the other hand goals are negotiated by members and membership is self-selecting, by identifying with the domain of knowledge of the CoP. Knowledge networks are at the other end of the spectrum to work groups and are an informal set of relationships which are primarily concerned with passing on knowledge, rather than the full range of knowledge management. In comparison, CoPs have a shared mission and desire in its members to work together to deepen their knowledge. CoPs also focus on the creation, storage, and utilisation of knowledge.

This review had a number of limitations. Only studies that directly had mention of a community of practice in the title, abstract, or full text were included. This meant that the diverse array of names used to refer to the concept of CoPs, such as situated learning, learning network, or even just community, were not included potentially excluding valuable studies. However, these phrases are used too ubiquitously in the field of healthcare, and as such, so broad a search was not feasible and so the search was focussed solely on the term community/ies of practice. Studies regarding medical education were also excluded, as has similarly been done in previous reviews [63], as these participants wouldn't necessarily be involved in providing healthcare directly. However, these studies may still have provided insights into the barriers and facilitators of engagement with healthcare themed CoPs. This review also did not employ a snowballing technique to examine the full list of references in each included study to broaden the search methodology. It was also not technically possible to carry out a logistic regression looking for the factors that were associated with effective CoPs as only 1 study reported a negative result.

CoPs in healthcare are aimed at improving clinical outcomes and have been shown to be effective. There is still progress to be made and a need for further studies with more rigorous methodologies, such as RCTs, to provide further support of the causality of CoPs on outcomes. As healthcare systems continue through their digital transformation journeys and healthcare workers have to manage an ever-mounting amount of knowledge, vCoPs in particular offer a method for improving outcomes and sharing vital information across an ever more complex healthcare landscape.

## Supporting information

**S1 Checklist. PRISMA 2020 checklist.**
(DOCX)

## Author Contributions

**Conceptualization:** Alexander P. Noar, Usman Jaffer.

**Data curation:** Alexander P. Noar, Hariharan Subbiah Ponniah.

**Formal analysis:** Alexander P. Noar, Hannah E. Jeffery.

**Investigation:** Alexander P. Noar, Hannah E. Jeffery, Hariharan Subbiah Ponniah, Usman Jaffer.

**Methodology:** Alexander P. Noar, Usman Jaffer.

**Project administration:** Alexander P. Noar, Usman Jaffer.

**Resources:** Alexander P. Noar.

**Software:** Alexander P. Noar.

**Supervision:** Alexander P. Noar, Usman Jaffer.

**Visualization:** Alexander P. Noar, Hannah E. Jeffery, Hariharan Subbiah Ponniah.

**Writing – original draft:** Alexander P. Noar, Hannah E. Jeffery, Hariharan Subbiah Ponniah, Usman Jaffer.

**Writing – review & editing:** Alexander P. Noar, Hannah E. Jeffery, Hariharan Subbiah Ponniah, Usman Jaffer.

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
