## [Decision Letter · Decision Letter 0]

19 Jul 2023

PONE-D-23-17541The Aims and Effectiveness of Communities of Practice in Healthcare: A Systematic ReviewPLOS ONE

Dear Dr. Noar,

Thank you for submitting your manuscript to PLOS ONE. After careful consideration, we feel that it has merit but does not fully meet PLOS ONE’s publication criteria as it currently stands. Therefore, we invite you to submit a revised version of the manuscript that addresses the points raised during the review process.

We look forward to receiving your revised manuscript.

Kind regards,

Edward Adekola Oladele, MD, MPH, PhD

Academic Editor

PLOS ONE

Journal Requirements:

"Usman Jaffer is founder and CEO of health-shared.com which is a platform that hosts

communities of practice for a number of healthcare related issues." 

3. We note that Figure 2 in your submission contain copyrighted images. All PLOS content is published under the Creative Commons Attribution License (CC BY 4.0), which means that the manuscript, images, and Supporting Information files will be freely available online, and any third party is permitted to access, download, copy, distribute, and use these materials in any way, even commercially, with proper attribution. For more information, see our copyright guidelines: http://journals.plos.org/plosone/s/licenses-and-copyright.

4. Please include a new copy of Table 1 and 2 in your manuscript; the current table is difficult to read. Please follow the link for more information: https://blogs.plos.org/plos/2019/06/looking-good-tips-for-creating-your-plos-figures-graphics/

Reviewers' comments:

Reviewer's Responses to Questions

**Comments to the Author**

1. Is the manuscript technically sound, and do the data support the conclusions?

Reviewer #1: Yes

Reviewer #2: Yes

2. Has the statistical analysis been performed appropriately and rigorously? 

Reviewer #1: Yes

Reviewer #2: Yes

3. Have the authors made all data underlying the findings in their manuscript fully available?

Reviewer #1: Yes

Reviewer #2: Yes

4. Is the manuscript presented in an intelligible fashion and written in standard English?

Reviewer #1: Yes

Reviewer #2: Yes

5. Review Comments to the Author

Reviewer #1: Dear Authors,

Below is my review of the Manuscript “The Aims and Effectiveness of Communities of Practice in Healthcare: A Systematic Review”.

Overall, the study presents a fairly comprehensive analysis of healthcare communities of practice. I have evaluated the manuscript based on its research design, methodology, results, and overall contribution to the field.

Materials and Methods

The search approach for primary research studies was quite broad from 1950 to 2022 and this approach was appropriate for the subject matter. However, it would be valuable to include detailed search strategies undertaken to include more geographies, and any differences observed in CoPs outcomes. It would also have been useful to present the framework for the review in more detail at this sage

Results

The results section presented an in-depth and well laid out overview of the findings.

Discussion

The discussion section expanded to provide a deeper analysis of the results and the authors clearly highlighted the limitations of their search strategy and the implications on “excluding valuable studies”. The reader may appreciate a brief comparison of CoPs with systematic of other knowledge sharing approaches relevant to the healthcare sector.

Minor Comments

1. Language and Grammar:

Abstract: Spelling error noted

In conclusion, this manuscript contributes to the understanding of communities of practice and its impact on knowledge sharing and collaboration.

Reviewer #2: Title: The Aims and Effectiveness of Communities of Practice in Healthcare: A Systematic Review

Comments

General comments:

This review is timely, and the authors should be congratulated for paying attention to the use of communities of practice, especially given their role during and now after covid to share information. The paper is well written.

Specific comments:

Title: The title “The Aims and Effectiveness of Communities of Practice in Healthcare: A Systematic Review” is appropriate for the paper

Abstract: The abstract is generally well written. There is a typo “hage” instead of “have”. Despite not being unstructured, the abstract does not clearly present the results. The last paragraph also seems to combine some statements that are part of background with a conclusion and implications or recommendations. For clarity, the authors could clearly articulate the findings, a conclusion, and some recommendations while still following the journals non-structured abstract policy.

Background: Well written and provides the context and justification for the review

Methods: Concise and easy to follow. Was there any limit to the geographic scope of the review. If so please clarify especially as results were presented by country where the research was conducted or the author’s home county.

Results: Well arranged and easy to follow. Table 1 cannot be read and could be added as supplemental material. All other tables are clear and well labeled.

Discussion and conclusion. Very well written and discuss the results comprehensively. Implications are well articulated.

6. PLOS authors have the option to publish the peer review history of their article (what does this mean?). If published, this will include your full peer review and any attached files.

Reviewer #1: No

Reviewer #2: No

---

## [Author Response · Author response to Decision Letter 0]

26 Aug 2023

Responses to editor:

PONE-D-23-17541

The Aims and Effectiveness of Communities of Practice in Healthcare: A Systematic Review

Dear Dr Edward Adekola Oladele,

Thank you very much for the reviewer’s comments and for inviting us to submit a revised manuscript. Please find our responses below. In the attached document, our responses to the questions are in blue, whilst changes to the text are given in green. This process has helped to significantly improve the manuscript. We look forward to hearing back from you.

Yours Sincerely, 

Alexander Noar

 

Journal Requirements:

Thank you for bringing our attention to this. The manuscript has been amended in line with these guidelines.

"Usman Jaffer is founder and CEO of health-shared.com which is a platform that hosts

communities of practice for a number of healthcare related issues." 

Thank you for bringing our attention to this. The manuscript and cover letter have been amended to include this statement.

3. We note that Figure 2 in your submission contain copyrighted images. All PLOS content is published under the Creative Commons Attribution License (CC BY 4.0), which means that the manuscript, images, and Supporting Information files will be freely available online, and any third party is permitted to access, download, copy, distribute, and use these materials in any way, even commercially, with proper attribution. For more information, see our copyright guidelines: http://journals.plos.org/plosone/s/licenses-and-copyright.

Thank you for bringing our attention to this. We were unfortunately not able to obtain the copyright permission for the images in Figure 2 so we have removed it from the manuscript. 

4. Please include a new copy of Table 1 and 2 in your manuscript; the current table is difficult to read. Please follow the link for more information: https://blogs.plos.org/plos/2019/06/looking-good-tips-for-creating-your-plos-figures-graphics/

Thank you for bringing our attention to this. New editable copies of Table 1 and 2 have been included in the manuscript in line with PLOS guidelines.

Thank you for bringing our attention to this. References have been checked and are in line with PLOS guidelines.

Response to Reviewer #1

Materials and Methods

The search approach for primary research studies was quite broad from 1950 to 2022 and this approach was appropriate for the subject matter. However, it would be valuable to include detailed search strategies undertaken to include more geographies, and any differences observed in CoPs outcomes. It would also have been useful to present the framework for the review in more detail at this sage

Thank you for this comment. There was no restriction on geography for the search. A description of differences in observed CoP outcomes between countries has also been added to the manuscript 

There was no restriction on geographical location

Outcomes varied across geographical location with North American countries such as Canada (91.7%) and USA (80%) having a higher percentage of studies with a quantitative element to their outcomes, in comparison to the UK (12.5%) or Australia (50%).

Discussion

The discussion section expanded to provide a deeper analysis of the results and the authors clearly highlighted the limitations of their search strategy and the implications on “excluding valuable studies”. The reader may appreciate a brief comparison of CoPs with systematic of other knowledge sharing approaches relevant to the healthcare sector.

Thank you for this comment. The discussion has been updated to include a comparison of CoPs to other knowledge management groups. 

CoPs differ from other knowledge management strategies such as work groups or knowledge networks. In work groups goals are pre-determined by a manager and members are usually assigned or selected by a leader. CoPs on the other hand goals are negotiated by members and membership is self-selecting, by identifying with the domain of knowledge of the CoP. Knowledge networks are at the other end of the spectrum to work groups and are an informal set of relationships which are primarily concerned with passing on knowledge, rather than the full range of knowledge management. In comparison, CoPs have a shared mission and desire in its members to work together to deepen their knowledge. CoPs also focus on the creation, storage, and utilisation of knowledge.

Minor Comments

1. Language and Grammar:

Abstract: Spelling error noted

Thank you for this comment. The spelling error in the abstract has now been corrected. 

have

Response to Reviewer #2

Abstract: The abstract is generally well written. There is a typo “hage” instead of “have”. Despite not being unstructured, the abstract does not clearly present the results. The last paragraph also seems to combine some statements that are part of background with a conclusion and implications or recommendations. For clarity, the authors could clearly articulate the findings, a conclusion, and some recommendations while still following the journals non-structured abstract policy.

Thank you for this comment. The abstract has been updated in line with your recommendations.

Methods: Concise and easy to follow. Was there any limit to the geographic scope of the review. If so please clarify especially as results were presented by country where the research was conducted or the author’s home county.

Thank you for this comment. There was no restriction on geography for the search. The manuscript has been updated to reflect this. 

There was no restriction on geographical location

Results: Well arranged and easy to follow. Table 1 cannot be read and could be added as supplemental material. All other tables are clear and well labeled

Thank you for this comment. Table 1 has been converted into an easier to read format.

---

## [Decision Letter · Decision Letter 1]

20 Sep 2023

The Aims and Effectiveness of Communities of Practice in Healthcare: A Systematic Review

PONE-D-23-17541R1

Dear Dr. Alexander Noar,

We’re pleased to inform you that your manuscript has been judged scientifically suitable for publication and will be formally accepted for publication once it meets all outstanding technical requirements.

Kind regards,

Tadashi Ito

Academic Editor

PLOS ONE

Additional Editor Comments (optional):

Reviewers' comments:

Reviewer's Responses to Questions

**Comments to the Author**

1. If the authors have adequately addressed your comments raised in a previous round of review and you feel that this manuscript is now acceptable for publication, you may indicate that here to bypass the “Comments to the Author” section, enter your conflict of interest statement in the “Confidential to Editor” section, and submit your "Accept" recommendation.

Reviewer #2: All comments have been addressed

2. Is the manuscript technically sound, and do the data support the conclusions?

Reviewer #2: Yes

3. Has the statistical analysis been performed appropriately and rigorously? 

Reviewer #2: Yes

4. Have the authors made all data underlying the findings in their manuscript fully available?

Reviewer #2: Yes

5. Is the manuscript presented in an intelligible fashion and written in standard English?

Reviewer #2: Yes

6. Review Comments to the Author

Reviewer #2: Thanks for submitting the revised manuscript which addresses the the reviewer comments. The manuscript is well written and adds important information for readers. The authors can be congratulated for addressing a topic important to knowledge management and use of information for decision making.

7. PLOS authors have the option to publish the peer review history of their article (what does this mean?). If published, this will include your full peer review and any attached files.

Reviewer #2: No

---

## [Editor Report · Acceptance letter]

2 Oct 2023

PONE-D-23-17541R1 

The Aims and Effectiveness of Communities of Practice in Healthcare: A Systematic Review 

Dear Dr. Noar:

I'm pleased to inform you that your manuscript has been deemed suitable for publication in PLOS ONE. Congratulations! Your manuscript is now with our production department. 

Kind regards, 

on behalf of

Dr. Tadashi Ito 

Academic Editor

PLOS ONE